# Choosing what we like vs liking what we choose: How choice-induced preference change might actually be instrumental to decision-making

**Douglas Lee**[1,2,3], **Jean Daunizeau**[1,2,3]*

**1** Sorbonne University, Paris, France, **2** Institut du Cerveau et de la Moelle épinière, Paris, France, **3** INSERM UMRS 1127, Paris, France

* jean.daunizeau@gmail.com

**Data Availability Statement:** All relevant data are within the paper and its Supporting Information files.

## Abstract

For more than 60 years, it has been known that people report higher (lower) subjective values for items after having selected (rejected) them during a choice task. This phenomenon is coined "choice-induced preference change" or CIPC, and its established interpretation is that of "cognitive dissonance" theory. In brief, if people feel uneasy about their choice, they later convince themselves, albeit not always consciously, that the chosen (rejected) item was actually better (worse) than they had originally estimated. While this might make sense from an intuitive psychological standpoint, it is challenging from a theoretical evolutionary perspective. This is because such a cognitive mechanism might yield irrational biases, whose adaptive fitness would be unclear. In this work, we consider an alternative possibility, namely that CIPC is -at least partially- due to the refinement of option value representations that occurs while people are pondering about choice options. For example, contemplating competing possibilities during a choice may highlight aspects of the alternative options that were not considered before. In the context of difficult decisions, this would enable people to reassess option values until they reach a satisfactory level of confidence. This makes CIPC the epiphenomenal outcome of a cognitive process that is instrumental to the decision. Critically, our hypothesis implies novel predictions about how observed CIPC should relate to two specific meta-cognitive processes, namely: choice confidence and subjective certainty regarding pre-choice value judgments. We test these predictions in a behavioral experiment where participants rate the subjective value of food items both before and after choosing between equally valued items; we augment this traditional design with both reports of choice confidence and subjective certainty about value judgments. The results confirm our predictions and provide evidence that many quantitative features of CIPC (in particular: its relationship with meta-cognitive judgments) may be explained without ever invoking post-choice cognitive dissonance reduction explanation. We then discuss the relevance of our work in the context of the existing debate regarding the putative cognitive mechanisms underlying CIPC.

**Funding:** DL received a PhD fellowship from the Labex Bio-Psy (Laboratory of excellence of Biology for Psychiatry). https://biopsy.fr/index.php/en/ The funder had no role in study design, data collection and analysis, decision to publish, or preparation of the manuscript.

**Competing interests:** The authors have declared that no competing interests exist.

# Introduction

The causal relationship between choices and subjective values goes both ways. By definition, choices are overt expressions of subjective values, which is the basis of decision theory [1]. However, one's choices also influence one's values, such that actions or items seem to acquire value simply because one has chosen them. Such "choice-induced preference change" (CIPC) has been repeatedly demonstrated via the so-called "free-choice paradigm" [2]. Here, people rate the pleasantness of (e.g., food) items both before and after choosing between pairs of equally pleasant items. Results show that the post-choice pleasantness ratings of chosen (rejected) options are typically higher (lower) than their pre-choice pleasantness ratings, which has traditionally been taken as empirical evidence for the existence of a "cognitive dissonance" reduction mechanism, triggered by difficult choices [3,4]. For example, people may rationalize their choice *ex post facto* as they think along the lines of, "I chose (rejected) this item, so I must have liked it better (worse) than the other one," and hence adjust their internal values accordingly [5]. Over the past decade, neuroimaging studies have demonstrated that the act of choosing between similarly-valued options causes changes in the brain's encoding of subjective values [6,7]. This has lent neurobiological support to the theory, and cognitive dissonance reduction is now the popular explanation behind a broad variety of important irrational sociopsychological phenomena, ranging from, for example, post-vote political opinion changes [8] to post-violence hostile attitude worsening [9].

This is not to say, however, that the theory of cognitive dissonance reduction has remained unchallenged. The first issue is theoretical in essence. In brief, it is unclear why evolutionary pressure would have favored post-choice cognitive dissonance reduction mechanisms, given that they could eventually induce irrational cognitive biases that have no apparent adaptive fitness [10–12]. For example, in the context of evidence-based decision making, standard cognitive dissonance theory predicts the appearance of confirmation and overconfidence biases. This is simply because weak beliefs should be reinforced by subsequent choices, despite the lack of any additional piece of evidence [13,14]. At least in principle, cognitive dissonance reduction may of course have other behavioral consequences that would overcompensate for the adverse selective pressure on cognitive biases [15–17]. Now, if it possessed such adaptive fitness, then it would undoubtedly be expressed in many other animal species. This is, however, an unresolved issue in the existing ethological literature [18,19]. Second, the main experimental demonstration of cognitive dissonance has also been challenged on statistical grounds. In 2010, Chen and Risen reported a methodological issue in the way CIPC had typically been measured and explained. The basic idea was that simple random variability in repeated value ratings could confound classical measures of CIPC in the context of the free-choice paradigm. The authors provided a detailed mathematical explanation for how such a statistical confound might eventually cause an apparent CIPC [20], and introduced a clever control condition. Here, both first and second value ratings are provided before any choice is ever made, thus precluding choice from causally influencing reported subjective values. Results show that significant CIPC occurs regardless of whether the choice is made before or after the second rating. Although this supports the validity of the authors' statistical criticism, subsequent studies also demonstrated that the magnitude of CIPC is significantly greater when the choice is made before the second value rating [21–23]. Taken together, the current theoretical and empirical bases of CIPC do not yet provide a straightforward portrait of why and how choice may influence subjective values.

Interestingly, recent neuroimaging evidence suggests that, in the context of typical free-choice paradigms, preference changes occur during the decision, not after it [24,25]. This is at

odds with the classical post-choice cognitive dissonance reduction theory. Recall that people are reluctant to make a choice that they are not confident about [26]. But contemplating competing possibilities during a choice provides a new context that highlights the unique aspects of the alternative options [27–29]. In turn, the act of choosing may change preferences by reappraising aspects of choice options that may not have been considered thoroughly before [30]. When faced with a difficult decision, this may enable people to reassess option values until they reach a satisfactory level of confidence [31–33]. This is important, because it allows for the possibility that preference changes may be instrumental for the process of decision making, which would resolve most theoretical concerns. This is the essence of our working hypothesis. We reason that decision difficulty drives people to reassess the values of the alternative options before committing to a particular choice. The ensuing refinement of internal value representations will eventually raise choice confidence enough to trigger the decision, which may or may not be aligned with pre-choice value ratings. Critically for our theory, the more difficult the decision, the more deliberation and potential reassessment of value representations, the more likely a change of mind and the related CIPC. In brief, post-choice cognitive dissonance reduction theory (hereafter: post-choice CDRT) states that people come to like what they have chosen. We rather suggest, somewhat trivially, that they may simply be choosing what they have come to like.

Importantly, our working hypothesis makes two original predictions that deviate from standard post-choice CDRT. Recall that the magnitude of CIPC is known to increase with the absolute difference between pre-choice option values, which is typically taken as a proxy for choice difficulty [4,23]. We argue, however, that choice difficulty is better defined in terms of the similarity of value representations. The critical difference is that value representations may be uncertain, i.e. peoples' feeling of liking and/or wanting a given choice option may be imprecise. In other terms, subjective estimates of choice difficulty derive from both value difference and metacognitive judgments about value uncertainty. Our first prediction regards the impact of the latter. We note that pre-choice value uncertainty is collected for each item, and is different from choice confidence (which we collect for each choice). In particular, we predict that CIPC should *decrease* with pre-choice value certainty. This is because pre-choice value certainty lowers choice difficulty, which de-motivates value reassessment. The prediction of standard post-choice CDRT regarding the impact of pre-choice value certainty is less clear. But if anything, we would argue that post-choice dissonance should be highest when pre-choice values were certainly equal, i.e. for choices with minimal value distance and maximal value certainty. Thus, under post-choice CDRT, CIPC should rather *increase* with pre-choice value certainty. Second, we predict that CIPC should *positively* correlate with choice confidence, controlling for the impact of decision difficulty. This is because, under our hypothesis, CIPC indirectly signals a successful improvement of choice confidence, due to reassessed values spreading apart. In contrast, post-choice CDRT would posit that choices made with low confidence should trigger the strong aversive dissonance feelings that eventually lead to CIPC. Thus, under post-choice CDRT, CIPC should *decrease* with choice confidence. These predictions are summarized in Fig 1 below.

That CDRT predicts the above three-way relationship between CIPC, choice confidence and value uncertainty, was, to the best of our knowledge, never stated until now. It is thus legitimate to criticize this prediction. We will briefly comment on this in the Discussion section. At this point, suffice it to say the primary goal of this study is to ask whether CIPC can be explained without invoking CDRT. We reason that it would be the case, should CIPC vary according to the predictions made under our working (alternative) hypothesis.

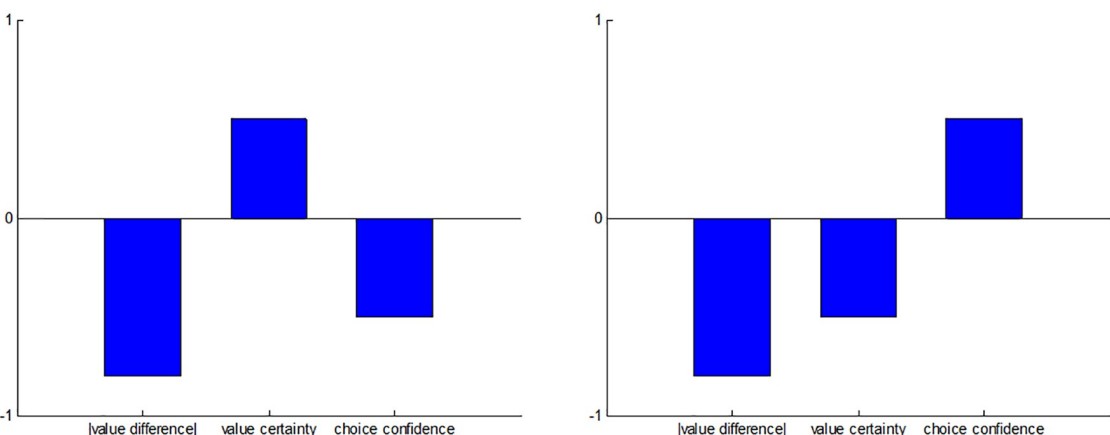

**Fig 1. Comparison of predicted relationships between meta-cognitive judgments and CIPC.** Here, we summarize the main predictions of how CIPC relate to meta-cognitive judgments under post-choice CDRT (left panel), and under our hypothesis (right panel). Blue bars depict putative partial correlations between CIPC on the one hand, and absolute pre-choice value difference (between options), pre-choice value certainty (averaged across both options), and choice confidence, on the other hand. One can see that, except for the effect of absolute value difference, our hypothesis make qualitatively distinct predictions from post-choice CDRT.

## Methods

### Ethics statement

Our analysis involved de-identified participant data and was approved by the ethics committee of the Institut du Cerveau et de la Moelle épinière (Paris, France). In accordance with the Helsinki declaration, all participants gave informed consent prior to commencing the experiment.

### Overview

We adapted the experimental design of Chen and Risen (2010), which includes two groups of participants. The so-called RCR (Rating, Choice, Rating) group of participants was asked to rate the value of a series of items both before and after making choices from pairs of items. In contrast, the RRC (Rating, Rating, Choice) group of participants rated the items twice before making the choices. As we will see, comparisons between the RCR and the RRC (control) group serve to rule out variants of Chen and Risen's statistical confound. In our adapted experimental design, when evaluating subjective values, participants now also rated their subjective certainty regarding their value judgment. In addition, they also reported how confident they were when making their choices. This allows us to assess the impact of both subjective uncertainty about value judgments and choice confidence on CIPC.

### Participants

A total of 123 people participated in this study. The RCR group included 65 people (45 female; age: mean = 29, stdev = 9, min = 19, max = 53). The RRC group included 58 people (34 female; age: mean = 33, stdev = 11, min = 18, max = 55). These groups did not show any significant difference in terms of age (p = 0.06) or gender (p = .27). All participants were native French speakers. Each participant was paid a flat rate of 12€ as compensation for one hour of time.

## Materials

We wrote our experiment in Matlab, using the Psychophysics Toolbox extensions [34]. The experimental stimuli consisted of 108 digital images, each representing a distinct sweet snack item (including cookies, candies, and chocolates). Prior to the experiment, participants received written instructions about the sequence of tasks, including typical visual examples of rating and choice trials.

## Experimental design

The experiment was divided into three sections, following the classic Free-Choice Paradigm protocol: Rating #1, Choice, Rating #2 (RCR group) or Rating #1, Rating #2, Choice (RRC group). Note that only in the RCR group do Rating #1 and Rating #2 correspond to pre-choice and post-choice ratings, respectively. Participants underwent a brief training session prior to the main testing phase of the experiment. There was no time limit for the overall experiment, nor for the different sections, nor for the individual trials. Within-trial event sequences are described below (see Fig 2 below).

**Rating.** Participants rated the stimulus items in terms of how much each item pleased them. The entire set of stimuli was presented to each participant, one at a time, in a random sequence (randomized across participants). At the onset of each trial, a fixation cross appeared at the center of the screen for 750ms. Next, a solitary image of a food item appeared at the center of the screen. Participants had to respond to the question, "Does this please you?" using a horizontal Likert scale (from "not at all" to "immensely") to report their subjective valuation of the item. Participants then had to respond to the question, "Are you sure?" using a vertical Likert scale (from "not at all" to "immensely") to indicate their level of subjective uncertainty regarding the preceding value judgment. We refer to the latter as a *value uncertainty rating* (which is not to be confounded with *choice confidence ratings*, see below). At that time, the next trial began.

**Choice.** Participants chose between pairs of items in terms of which item they preferred. The entire set of stimuli was presented to each participant, one pair at a time, in a random sequence of pairs. Each item appeared in only one pair. The algorithm that created the choice

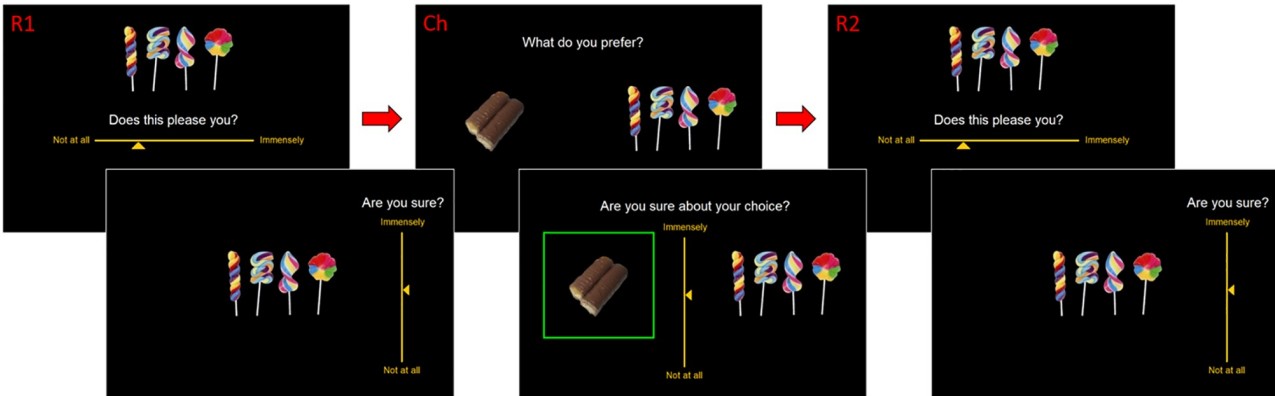

**Fig 2. Behavioural paradigm (RCR group).** The experiment was divided into three sessions. First, participants were asked to rate each item's value and report their level of uncertainty regarding this value rating. In the main text, we refer to these as "pre-choice value" and "pre-choice value certainty", respectively. Second, participants are asked to chose between pair of items and report their confidence about this decision. In the main text, we refer to the latter as "choice confidence". The third session is identical to the first one, and we refer to the corresponding measures as "post-choice value" and "post-choice value certainty".

pairs first sorted all items into 10 bins, then paired off (at least) half of the items within each bin, then paired off all remaining items across bins. This ensured that at least half of choices would be between items of similar subjective value (value rating difference < 1/10 of the full rating scale, as shown in previous studies to cause CIPC), but that a substantial portion would be associated with greater value differences. At the onset of each trial, a fixation cross appeared at the center of the screen for 750ms. Next, two images of snack items appeared on the screen: one towards the left and one towards the right. Participants had to respond to the question, "What do you prefer?" using the left or right arrow key. Participants then had to respond to the question, "Are you sure about your choice?" using a vertical Likert scale to report their level of confidence in the preceding choice. We refer to the latter as a *choice confidence rating* (not to be confounded with *value uncertainty ratings*, see above). At that time, the next trial began.

## Results

Before testing our hypothesis (against both statistical confounds and standard post-choice cognitive dissonance reduction theory), we performed a number of simple data quality checks. First, we assessed the test-retest reliability of both value judgments and their associated value certainty reports. For each participant, we thus measured the correlation between ratings #1 and #2 (across items). We found that both ratings were significantly reproducible (value ratings: correlation = 0.862, 95% CI [0.838, 0.886], p<0.001; value certainty ratings: correlation = 0.472, 95% CI [0.409, 0.535], p<0.001). Second, we asked whether choices were consistent with value ratings #1. For each participant, we thus performed a logistic regression of paired choices against the difference in pre-choice value ratings. We found that the balanced prediction accuracy was beyond chance level (prediction accuracy given pre-choice ratings = 0.685, 95% CI [0.666, 0.703], p<0.001). Third, we checked that choice confidence increases both with the absolute value difference between paired items, and with the average value certainty rating (of the paired items). For each participant, we thus performed a multiple linear regression of choice confidence against pre-choice absolute value difference and pre-choice mean value certainty (ratings #1). A random effect analysis shows that both have a significant effect at the group level ($R^2$ = 0.223, 95% CI [0.188, 0.258]; absolute value difference: GLM beta = 9.046, 95% one-sided CI [7.841, $\infty$], p<0.001; mean value certainty: GLM beta = 3.157, 95% one-sided CI [2.137, $\infty$], p<0.001). Fourth, we asked whether we could reproduce previous findings that CIPC is higher in the RCR group than in the RRC group. For each participant, we thus measured the magnitude of CIPC in terms of the so-called "spreading of alternatives" (SoA), calculated as the mean difference in value rating gains between chosen and unchosen items (SoA = [rating#2-rating#1]$_{chosen}$—[rating#2-rating#1]$_{unchosen}$). As expected, we found that SoA is significant in both groups (RCR group: SoA = 5.033, 95% CI [4.118, 5.949], p<0.001; RRC group: SoA = 2.635, 95% CI [2.047, 3.224], p<0.001). In addition, SoA is significantly higher in the RCR group than in the RRC group (SoA difference = 2.398, p<0.001) (Fig 3).

In what follows, and unless stated otherwise, we focus on the RCR group of participants. Recall that, under our hypothesis, the deliberation that takes place during the decision process is expected to cause a refinement of internal value representations until a target level of choice confidence is met and the decider commits to a choice. To begin with, we thus asked whether certainty about value judgments improved after the choice had been made. For each participant, we thus estimated the mean difference between post-choice and pre-choice value certainty ratings (across all items). A random effect analysis then shows that post-choice value certainty ratings are significantly higher than pre-choice value certainty ratings (value certainty

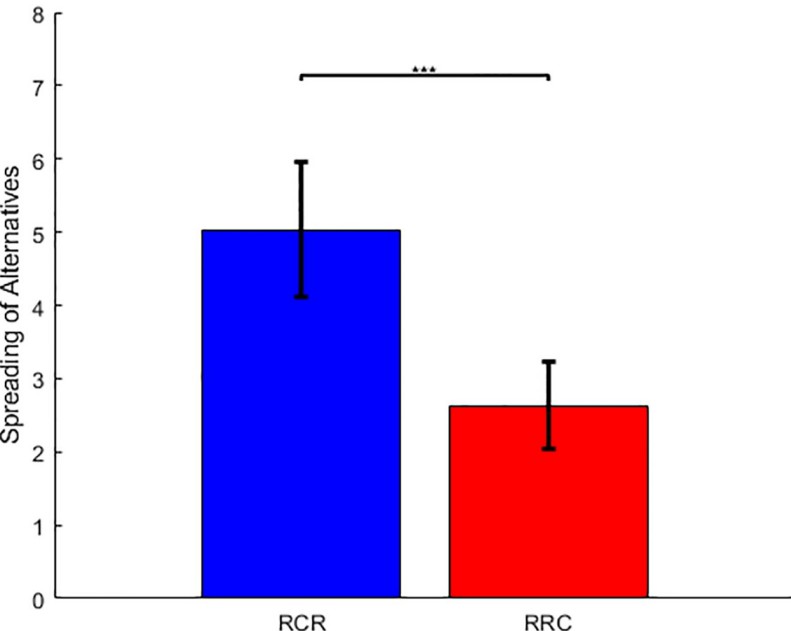

**Fig 3. Comparison of RCR and RRC groups.** The mean spreading of alternatives (SoA = [rating#2-rating#1]$_{chosen}$—[rating#2-rating#1]$_{unchosen}$) is shown for both the RCR (blue) and the RRC (red) group. Error bars depict 95% confidence intervals. This reproduces the results of Chen and Risen (2010).

increase = 3.781, 95% CI [1.810, 5.752], p<0.001). This finding supports our claim but does not provide evidence for or against classical post-choice cognitive dissonance reduction theory. We then asked whether post-choice ratings better predict choice (and choice confidence) than pre-choice ratings. First, we performed another logistic regression of choices, this time against the difference in *post*-choice value ratings (ratings #2). The ensuing balanced prediction accuracy is significantly higher than with pre-choice value ratings: prediction accuracy given post-choice ratings = 0.787, 95% CI [0.770, 0.804], prediction accuracy gain (post-pre) = 0.103, 95% CI [0.082, 0.124], p<0.001 (Fig 4). Second, we regressed choice confidence, this time against *post*-choice absolute value difference and mean value certainty. The ensuing amount of explained variance is higher than with pre-choice ratings: $R^2$ given post-choice ratings = 0.245, 95% CI [0.209, 0.281], $R^2$ gain (post-pre) = 0.022, 95% CI [0.001, 0.042], p = 0.02 (Fig 4). When testing for the significance of differences in pre-choice and post-choice regression parameters, we found that this gain in explanatory power is more likely to be due to value ratings (GLM beta difference = 0.763, 95% one-sided CI [0.180, ∞], p = 0.018) than to value certainty ratings (GLM beta difference = 0.158, 95% one-sided CI [-0.451, ∞], p = 0.34).

The above results are important, because they validate basic requirements of our working hypothesis. However, they are equally compatible with both CDRT and our working hypothesis. Thus, we now focus on testing the predicted three-way relationship between CIPC, value certainty and choice confidence, which discriminates the alternative mechanisms (cf. Fig 1 above). For each participant, we performed a multiple linear regression of SoA onto absolute difference in pre-choice value ratings (within choice pairs), mean pre-choice judgment certainty reports (within choice pairs), and choice confidence (Fig 5). As expected, a random effect analysis on the ensuing parameter estimates shows that SoA significantly decreases when the absolute difference in pre-choice value ratings increases (GLM beta = -0.295, 95% one-

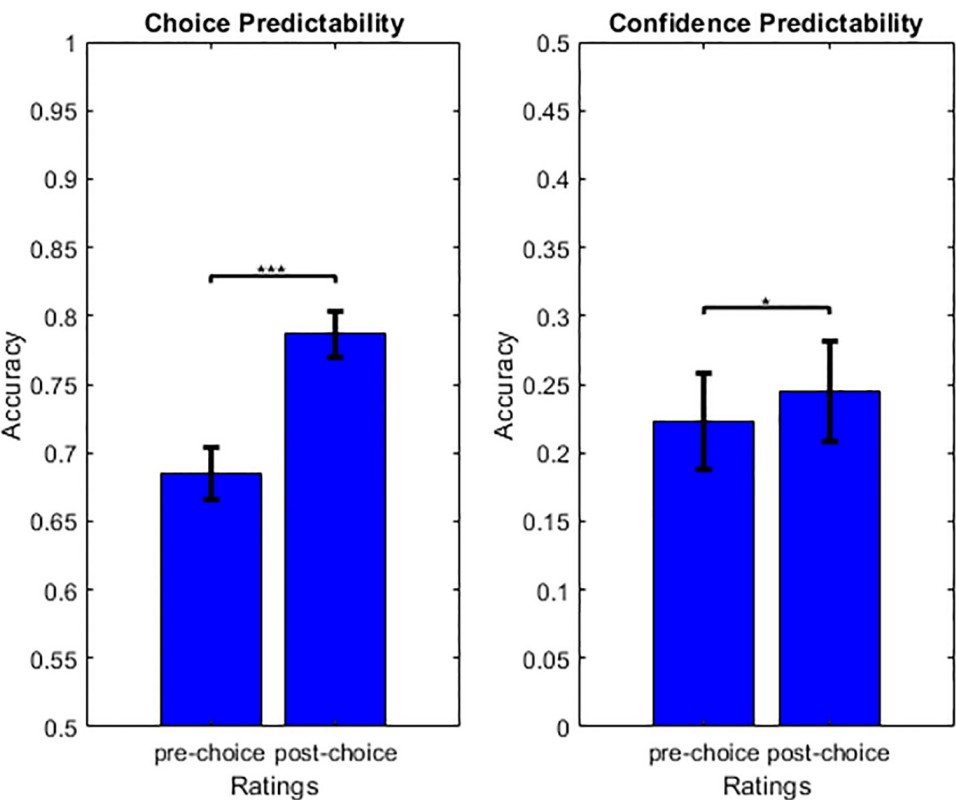

**Fig 4. Prediction accuracy of choice and choice confidence.** Left: Mean choice prediction accuracy is plotted for pre-choice (left) and post-choice (right) value ratings. Right: Mean choice confidence prediction accuracy is plotted for pre-choice (left) versus post-choice (right) value and value uncertainty ratings. Error bars depict 95% confidence intervals.

sided CI [-∞, -0.265], p<0.001). More importantly, we found that SoA significantly decreases when pre-choice value certainty ratings increases (GLM beta = -0.065, 95% one-sided CI [-∞, -0.036], p<0.001) and significantly increases when choice confidence increases (GLM beta = 0.200, 95% one-sided CI [0.170, ∞], p<0.001). The latter findings support our hypothesis, and are incompatible with classical post-choice CDRT.

Finally, we aimed at ruling out statistical confounds. This can be done by showing that if the above statistical relationships exist in the RRC group, they should be significantly weaker than in the RCR group. We thus performed the above analyses on data acquired in participants from the RRC group, which we compared to the RCR group of participants using standard random effect analyses. First, the gain in choice prediction accuracy (from rating #1 to rating #2) is significantly higher in the RCR group than in the RRC group (accuracy gain difference = 0.0362, p = 0.008). Second, and most importantly, both the impact of absolute pre-choice value difference (GLM beta diff = -0.095, p = 0.0039) and choice confidence (GLM beta diff = 0.044, p = 0.049) on SoA are significantly higher in the RCR than in the RRC group. Note that some comparisons between the two groups turned out not to be significant (gain in value certainty ratings: p = 0.17, gain in confidence prediction accuracy: p = 0.58, impact of value certainty on SoA: p = 0.096). Nevertheless, taken together, these findings are unlikely under a chance model of random variations in value ratings.

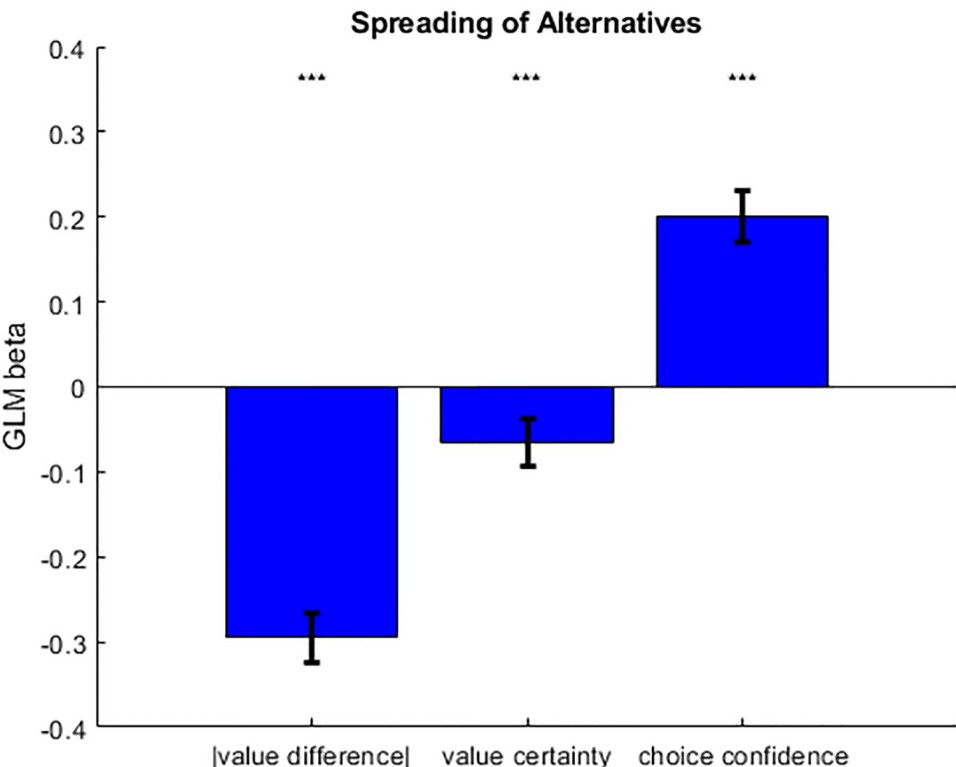

**Fig 5. Observed relationships between meta-cognitive judgments and CIPC.** This figure shows the results of a multiple linear regression of CIPC onto absolute value difference, value certainty and choice confidence, in terms of the mean standardized regression coefficients (error bars depict 95% confidence intervals). Note that regressors were orthogonalized in that order, i.e. rightmost bars show partial correlation coefficients (after having accounted for the effects depicted by the leftmost bars). Compare this figure to Fig 1 above.

## Discussion

In this work, we have presented empirical evidence that challenges standard interpretations of CIPC, in particular: post-choice cognitive dissonance reduction theory or CDRT (and its "self-perception" variants). According to standard post-choice CDRT, choices made with low confidence trigger strong aversive dissonance feelings that are resolved by retrospectively matching internal value representations to the choice. We would rather say that no choice commitment is made until internal value representation refinements allow choice confidence to reach a satisfying (non-aversive) level. Our results show that CIPC varies as predicted by the latter scenario. This is important, because this implies that CIPC may be explained without referring to CDRT.

In our experiment, the correlation between CIPC and RT turns out to be significant and negative, even when controlling for pre-choice decision variables (mean correlation = -0.1236, p<0.001). Can we interpret this result as providing evidence in favor or against our working assumption? Recall that we hypothesize that value reassessment during decision-making is a critical determinant of CIPC. This might tempt one to predict that longer RT should lead to greater CIPC. The logic of this idea would be that difficult decisions cause people to deliberate longer, which in turn allows for larger changes in value estimates. But in fact, predictions regarding the relationship between RT and CIPC actually depend upon two types of implicit assumptions. First, decision deliberation is likely to unfold along two dimensions, namely:

intensity and duration. From this perspective, trial-by-trial variations in RT are a good proxy for trial-by-trial variations in decision deliberation only if intensity is constant over trials. But this may not be the case. In particular, information processing rate may vary. If such variations in intensity overcompensate for variations in duration, then short RT would signal intense decision deliberation. Second, even if RT were a reliable empirical proxy for decision deliberation, the expected relationship between RT and CIPC would remain ambiguous. This is because it would still depend upon how decisions are triggered. Consider, for example, drift-diffusion decision models of value-based decision making [35,36]. Here, a decision is triggered when some decision variable (e.g., value difference) eventually reaches a collapsing bound. Importantly, those decisions that encounter little or no value spreading take more time because the system has to wait until the bound (i.e., the evidence that is demanded for triggering a decision) is small enough. In turn, choices with long RT would be those made with low confidence and small CIPC. The interested reader will find numerical simulations demonstrating this effect in the Supporting Information (S1 Text). Although the observed correlation between CIPC and RT corroborates this scenario, other alternative computational scenarios (that suggest different mechanisms for how decisions are triggered) may make qualitatively different predictions. For example, if bounds are not collapsing, then drift-diffusion models would predict no systematic relationship between RT and CIPC. Taken together, these issues suggest that the observed correlation between CIPC and RT cannot be taken as clear direct evidence that CIPC is driven by value reassessment that happens during the decision.

As a side note, we acknowledge that our group comparison (cf. RCR group versus RRC control group) suffers from the typical limitations of between-subject designs. In particular, and although we did not find any significant age difference between the two groups, trivial age differences may still in principle confound our comparison. If this was the case, then one may challenge the validity of our control RRC group. Recall that this control is important for demonstrating that CIPC exists above and beyond the expected statistical bias reported in Chen and Risen (2010). Here, we used the RRC control group to show that the magnitude of the statistical relationship between CIPC and metacognitve judgments (in the RCR group) exceeds what can be expected from a situation where decision processes cannot interfere (cf. RRC group). Given that virtually nothing is known about the effect of age on CIPC [37], it is thus theoretically possible that our observed group differences may in fact be driven by some variant of Chen and Risen's statistical artifact. However, Chen and Risen's reasoning does not apply straightforwardly to predictions regarding statistical relationships between CIPC and metacognitive judgments. In addition, such age confound is a priori unlikely, given that many studies previously already reported a significant difference between RRC and RCR groups or conditions [21–23,38].

One might challenge our interpretation of the observed relationship between confidence and CIPC, in terms of evidence against post-choice CDRT. For example, one might argue that CIPC could occur after people commit to a choice, but before they get a feel for how confident they are about that choice. This would seem paradoxical, however, in the sense that experiencing cognitive dissonance in this context simply means feeling uneasy about one's choice, i.e. lacking confidence about it. In any case, this line of reasoning cannot apply to the observed impact of value certainty on CIPC. Recall that we probe metacognitive judgments about value certainty before the choice, using rating scales at the time when each item is presented (immediately after first-order value judgments). Therefore, the relationship between CIPC and value certainty that we disclose empirically cannot derive from metacognitive processes that occur after the choice has been made. In any case, we are not trying to provide evidence for the fact that CIPC occurs during -as opposed to after- the decision (we discuss the related literature below). Rather, we are trying to provide evidence in favor of another interpretation of CIPC,

namely: that people reassess option values until they reach a satisfactory level of confidence (at which point they commit to a choice). Or more precisely, we are trying to show that CIPC is in fact expected, without referring to any post-choice cognitive dissonance reduction mechanism. It turns out that our working hypothesis makes two quantitative predictions that deviate from standard post-choice CDRT. These predictions regard the relationship between CIPC and meta-cognitive judgments. For the purpose of validating these predictions, and therefore showing that CIPC behaves as expected, these meta-cognitive measures are, in our opinion, reliable enough.

Let us now discuss our results in the context of the existing literature affording evidence in favor of post-choice CDRT. In particular, CIPC was recently shown to only occur for choices where the agent is later able to recall which option was chosen and which was rejected [21]. This is somewhat problematic because remembering choices has no causal role under our pre-choice value refinement hypothesis. This contrasts with standard CDRT, where post-choice option re-evaluation requires the memory trace of relevant choices. The latter interpretation is compatible with the observation that, when re-evaluating items after the choice, activity in the hippocampus discriminates between remembered and non-remembered choices [39]. These two studies thus provide apparent evidence for post-choice CDRT. However, we contend that this theory remains unsupported until empirical evidence is found for memory traces of information that is critical for post-choice CIPC (namely: whether an option was chosen or rejected, what was the option's pre-choice value, and which option was the alternative during the relevant choice). In addition, the relationship between CIPC and memory might be confounded by choice difficulty. In brief, the more difficult a decision is, the more value reassessment it will eventually trigger, the more likely the agent is to remember his/her choice. Alternatively, post-choice reports of internal values may rely on slightly unstable episodic memory traces of intra-choice CIPC. The latter scenario is actually compatible with the fact that activity in the left dorsolateral prefrontal cortex (during choice) predicts the magnitude of CIPC only when the choices are remembered [7], and also with the intra-choice CIPC interpretation of the causal impact of post-choice activity perturbations (see below). In any case, either or both of these scenarios would explain why intra-choice CIPC might exhibit an apparent (non-causal) relationship with choice memory. Finally, note that the causal implication of memory is inconsistent with the assessment of amnesic patients, who exhibit normal CIPC despite severe deficits in choice memory [40].

Now recent neuroimaging findings shed light on the question of whether CIPC occurs during or after the decision. This question is important, because evidence for a post-choice CIPC disconfirms our working hypothesis. On the one hand, a few brain stimulation studies suggest that perturbing brain activity *after the choice* (in particular: in the left dorsolateral and/or posterior medial frontal cortices) disrupts the observed CIPC [41,42]. Although compatible with post-choice CDRT, such causal effects can be due to the post-choice disturbance of value representations that resulted from intra-choice CIPC. On the other hand, many recent studies show that brain activity measured *during the choice process* is predictive of the magnitude of CIPC [7,24,25,43,44]. Unsurprisingly, key regions of the brain's valuation and cognitive control systems are involved, including: the right inferior frontal gyrus, the ventral striatum, the anterior insula and the anterior cingulate cortex (ACC). Note that current neurocomputational theories of ACC suggest that it is involved in controlling how much mental effort should be allocated to a given task, based upon the derivation of the so-called "expected value of control" [45,46]. This is highly compatible with our results, under the assumption that pre-choice value reassessment is a controlled and effortful process that trades mental effort against choice confidence [47]. We will pursue this computational scenario in subsequent publications.

Nevertheless, we consent that CIPC may be driven by both pre-choice value reassessment and post-choice cognitive dissonance reduction mechanisms. The quantitative contribution of the latter effect, however, may have been strongly overestimated. In our view, this is best demonstrated, though perhaps unintentionally, in the results of the "blind choice" study from Sharot and colleagues [48]. Here, participants rated items both before and after making a blind choice that could not be guided by pre-existing preferences (because the items were masked). Critically, although blind choice precludes any instrumental value refinement process, preferences were altered after the choice. Interestingly, the effect size is rather small, i.e. the ensuing CIPC magnitude was estimated to be around $0.07 \pm 0.03$. This is to be compared with the CIPC magnitude of the RCR and RRC conditions in two other studies by the same authors [23,30], namely: $0.38 \pm 0.08$ (RCR condition, with non-blind choices) and $0.11 \pm 0.06$ (RRC condition, which was not included in the "blind choice" study). In other terms, CIPC under blind choice is smaller than the apparent CIPC that unfolds from the known statistical confounds of the free choice paradigm. Note that if this had not have been the case, then post-choice CIPC cognitive reduction effects would dominate and we would not have confirmed our predictions.

## Conclusion

In conclusion, our results lend support to the hypothesis that choice-induced preference change is caused, at least partially, by an intra-choice refinement of option value representations that is motivated by difficult decisions. Such mechanism is qualitatively distinct from those considered in the context of post-choice cognitive dissonance resolution theories. We also demonstrate the relevance of meta-cognitive processes (cf. reports of choice confidence and certainty about value judgments) to choice-induced preference change. This contributes to moving forward the state of the 60-year-old research on the reciprocal influence between decision making and subjective values.

## Supporting information

**S1 Text. This note discusses the relationship between responses times (RT) and choice-induced preference changes (CIPC), from a computational standpoint.** In particular, this note summarizes numerical simulations performed under so-called drift-diffusion decision models.
(PDF)

**S1 Code.**
(ZIP)

## Author Contributions

**Conceptualization:** Douglas Lee, Jean Daunizeau.

**Formal analysis:** Douglas Lee, Jean Daunizeau.

**Funding acquisition:** Douglas Lee, Jean Daunizeau.

**Investigation:** Douglas Lee, Jean Daunizeau.

**Methodology:** Douglas Lee, Jean Daunizeau.

**Project administration:** Douglas Lee, Jean Daunizeau.

**Supervision:** Jean Daunizeau.

**Validation:** Douglas Lee, Jean Daunizeau.

**Writing – original draft:** Douglas Lee.

**Writing – review & editing:** Douglas Lee, Jean Daunizeau.

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
