## [Decision Letter · Decision Letter 0]

25 Jul 2019

PONE-D-19-17905

Choosing what we like vs liking what we choose: How choice-induced preference change might actually be instrumental to decision-making

PLOS ONE

Dear Dr. Daunizeau,

Thank you for submitting your manuscript to PLOS ONE. After careful consideration, we feel that it has merit but does not fully meet PLOS ONE’s publication criteria as it currently stands. Therefore, we invite you to submit a revised version of the manuscript that addresses the points raised during the review process.

Although reviewers appreciate the addressed question, both reviewers raise serious concerns about the main claim of the manuscript and whether and how this claim is supported by the experimental results. In addition, Reviewer # 2 points out important issues about the overall tone of the manuscript and implications of the findings. I think you should fix these issues or conduct additional experiments to support your claims.

We would appreciate receiving your revised manuscript by Sep 24 2019 11:59PM. To enhance the reproducibility of your results, we recommend that if applicable you deposit your laboratory protocols in protocols.io, where a protocol can be assigned its own identifier (DOI) such that it can be cited independently in the future. For instructions see: http://journals.plos.org/plosone/s/submission-guidelines#loc-laboratory-protocols

We look forward to receiving your revised manuscript.

Kind regards,

Alireza Soltani

Academic Editor

PLOS ONE

Journal Requirements:

2) Please ensure that you refer to Figure 4 in your text as, if accepted, production will need this reference to link the reader to the figure.

3) Please include captions for your Supporting Information files at the end of your manuscript, and update any in-text citations to match accordingly. Please see our Supporting Information guidelines for more information: http://journals.plos.org/plosone/s/supporting-information.

Reviewers' comments:

Reviewer's Responses to Questions

**Comments to the Author**

1. Is the manuscript technically sound, and do the data support the conclusions?

Reviewer #1: Partly

Reviewer #2: No

2. Has the statistical analysis been performed appropriately and rigorously? 

Reviewer #1: Yes

Reviewer #2: Yes

3. Have the authors made all data underlying the findings in their manuscript fully available?

Reviewer #1: No

Reviewer #2: Yes

4. Is the manuscript presented in an intelligible fashion and written in standard English?

Reviewer #1: No

Reviewer #2: Yes

5. Review Comments to the Author

Reviewer #1: This manuscript describes a study that purports to challenge the mechanistic basis of post-decision cognitive dissonance theory. The tone and language throughout the manuscript needs to be reigned in substantially – this was an incremental behavioral study that shows some evidence that value and choice confidence play a role in influencing changes in valuation quantified after decision-making – it is not a death knell to the theoretical postulates of post-decision cognitive dissonance theory. Some of the hypotheses are framed as novel, but as written, it is not clear that that is the case. Likewise, the discussion frames the report largely in the context of memory research, despite the fact that memory is not measured in the study. These and additional comments are provided below:

The use of parenthetical statements in the abstract and throughout the introduction is confusing.

The writing throughout has a tone that is quite unusual for an academic paper. Just a couple of examples from among: “We take inspiration from..”; “In essence, our experimental design is borrowed from...”

In the abstract and MS, the authors need to tone down their claims about whether or not CIPC is adaptive and the implications of CIPC from cognitive and evolutionary perspectives. This extends to claims that CIPC results in irrational biases etc.

The authors state that they have two hypotheses that are original predictions:

“First, the magnitude of CIPC should increase with decision difficulty.” This has been a hypothesis in the context of CIPC for quite sometime. That is why forced choices in the majority of CIPC studies are made between items that are rated as similarly attractive/valuable/desirable etc. At least as written, it is hard to see the distinction between subjective uncertainty and difficulty – they do not seem to be orthogonal factors, but confounded. E.g., This is difficult because I’m not sure which one I’m going to choose.

“In particular, we predict that CIPC should increase with subjective uncertainty regarding pre-choice value judgment. This goes against standard post-choice cognitive dissonance reduction theory, because post-choice dissonance should be highest when pre-choice values are similar, with a high judgment certainty.” This second statement doesn’t make sense. CIPC is typically larger when pre-choice values are similar and there is high UNcertainty. Choices are difficult because you are UNcertain which to choose. Prior work also demonstrates effects are greater when you value the items you are choosing about more. It is not clear what is new here.

Is there a significant difference in the age across the two groups? If so, all analyses must control for age.

There are infinity symbols in many of the confidence intervals.

“Recall that, under our hypothesis, effort allocation during the decision process is expected to refine internal value representations up until a target level of confidence is met and the decider commits to a choice.” Given that effort allocation is a construct of interest, response time should be reported and analyzed? Even this cannot determine that internal value representations are refined up until a target level of confidence is met. The authors need to reframe this construct in the context of what they are objectively measuring.

The discussion of memory steps beyond the data collected, yet seems to be the primary focus of the discussion. E.g., “Our results apparently contradict the recent finding that CIPC only occurs for choices where the agent is later able to recall which option was chosen and which was rejected (Salti et al, 2014).” The authors do not test memory, thus their results cannot contradict these findings. The majority of this entire paragraph seems inconsistent with the data that were actually collected.

Reviewer #2: Choice induced preference change (CIPC) is a fascinating subject as a behavioral manifestation of the fundamental process of decision making. The problem has been around for almost 60 years. Yet how this phenomenon occur remains a mystery. One of the important questions about CIPC is when the processes causing preference change occur. The authors tried to address this question with a new behavioral paradigm, which included confidence reports at preference rating and choice between rated objects. The authors’ hypothesis is that the metacognitive processes occurring during a decision making induce the preference change. I have a sympathy to this claim itself. But the evidence presented in this study cannot prove this hypothesis.

Major Point

The processes related to the confidence report can occur after the decision making. Thus, whatever the relationship between the confidence reports and CIPC cannot prove the author’s claim.

6. PLOS authors have the option to publish the peer review history of their article (what does this mean?). If published, this will include your full peer review and any attached files.

Reviewer #1: No

Reviewer #2: No

---

## [Author Response · Author response to Decision Letter 0]

25 Sep 2019

See our response letter (attached to the main mascnuript).

---

## [Decision Letter · Decision Letter 1]

22 Oct 2019

PONE-D-19-17905R1

Choosing what we like vs liking what we choose: How choice-induced preference change might actually be instrumental to decision-making

PLOS ONE

Dear Dr. Daunizeau,

Thank you for submitting your manuscript to PLOS ONE. After careful consideration, we have decided that your manuscript does not meet our criteria for publication and must therefore be rejected.

More specifically, both reviewers think that the current experiment cannot discriminate between mechanisms of pre-choice value refinements from post-choice cognitive dissonance reduction, which severely undermines the main claim of the manuscript. On the one hand, choice confidence, or certainty about choice measures, is obtained only after the choice is made and not during decision-making processes. On the other hand, it is not clear that value certainty, which is recorded before the choice, can serve as metacognitive measure of decision process before it actually happens. As mentioned by Reviewer # 1, reaction time date could provide some insights but unfortunately were not included in the manuscript.   

I am sorry that we cannot be more positive on this occasion, but hope that you appreciate the reasons for this decision.

Yours sincerely,

Alireza Soltani

Academic Editor

PLOS ONE

Reviewers' comments:

Reviewer's Responses to Questions

**Comments to the Author**

1. If the authors have adequately addressed your comments raised in a previous round of review and you feel that this manuscript is now acceptable for publication, you may indicate that here to bypass the “Comments to the Author” section, enter your conflict of interest statement in the “Confidential to Editor” section, and submit your "Accept" recommendation.

Reviewer #1: (No Response)

Reviewer #2: (No Response)

2. Is the manuscript technically sound, and do the data support the conclusions?

Reviewer #1: No

Reviewer #2: Yes

3. Has the statistical analysis been performed appropriately and rigorously? 

Reviewer #1: Yes

Reviewer #2: Yes

4. Have the authors made all data underlying the findings in their manuscript fully available?

Reviewer #1: Yes

Reviewer #2: Yes

5. Is the manuscript presented in an intelligible fashion and written in standard English?

Reviewer #1: Yes

Reviewer #2: Yes

6. Review Comments to the Author

Reviewer #1: The authors failed to address key issues raised in the prior review.

-From the most basic perspective, the methods utilized in this paper cannot discriminate mechanisms of pre-choice value refinements from post-choice cognitive dissonance reduction. Certainty about choice measures are obtained after the choice is made. There is no way to disentangle what happens during the decision to what happens after, as data about certainty were only collected after the decision. Moreover, the authors state: “Therefore, the relationship between CIPC and value certainty that we disclose empirically cannot derive from metacognitive processes that occur after the choice has been made.” As noted in the prior review, the authors have no data to support this and other assertions about the cognitive processes engaged during and after decision-making. Response times could perhaps provide some empirical support for their assertions, however, they failed to include these data in their revision.

- The authors contradict themselves when positing what standard theories predict. Here is one example: “In contrast, standard post-choice cognitive dissonance reduction theory predicts that post-choice dissonance should be highest when pre-choice values were a priori certainly equal, i.e. for choices with minimal value distance and maximal value certainty.” ... “In contrast, standard dissonance reduction theory would posit that choices made with low confidence should trigger the strong aversive dissonance feelings that eventually lead to CIPC.” First they state that CIPC would be highest when pre-choice are “certainly equal” – which I take to mean there is high confidence in their value. Second they say that CIPC would be highest when pre-choices are “made with low confidence” – which I take to mean there is low confidence in their value. These assertions continue to be problematic.

- The authors do not measure memory. As noted in the prior review, the paragraph claiming the results contradict recent memory-based findings steps well beyond the data. It should be removed.

- No statistics are provided regarding age and sex distributions across conditions.

- The authors fail to state that one-sided statistical tests were used in some analyses.

Reviewer #2: The authors addressed my concern about the confidence reports after the decision by stating that the other metacognitive measure, value certainty. Value certainty is recorded before the choice. But does this serve as metacognitive measure of intra-choice process? I commented on this measure (reports of choice confidence) because this is supposed to reflect the intra-choice process. The pre-choice measures of certainty about value judgments have limitations in estimating the intra-choice processes because the choice process has not happened yet. I recommend that the authors should be careful about the distinctions of the two metacognitive measures and clearly state the limitations.

7. PLOS authors have the option to publish the peer review history of their article (what does this mean?). If published, this will include your full peer review and any attached files.

Reviewer #1: No

Reviewer #2: No

- - - - -

---

## [Author Response · Author response to Decision Letter 1]

12 Nov 2019

[See submitted response letter]

---

## [Decision Letter · Decision Letter 2]

20 Jan 2020

PONE-D-19-17905R2

Choosing what we like vs liking what we choose: How choice-induced preference change might actually be instrumental to decision-making

PLOS ONE

Dear Dr. Daunizeau,

Thank you for submitting your revised manuscript to PLOS ONE. After careful consideration and consultation between editorial members, we feel that your manuscript has merit but does not fully meet PLOS ONE’s publication criteria as it currently stands. More specifically, as pointed out by Reviewer #1, RT data could provide essential information about proving or disproving alternative hypotheses/models and therefore, analyses of RT data should be provided and discussed. In case RT data were not collected, this should be mentioned as a limitation of the study. We also feel that some of the statements about debunking cognitive dissonance theory based on the results of one experiment should be toned down. 

**Therefore, we invite you to submit a revised version of the manuscript that fully addresses these and other last points raised by Reviewer #1, along with a response letter. Your revised manuscript and response letter will not be sent out to review again and instead, will be evaluated at the editorial level. **

We would appreciate receiving your revised manuscript by Mar 05 2020 11:59PM. To enhance the reproducibility of your results, we recommend that if applicable you deposit your laboratory protocols in protocols.io, where a protocol can be assigned its own identifier (DOI) such that it can be cited independently in the future. For instructions see: http://journals.plos.org/plosone/s/submission-guidelines#loc-laboratory-protocols

We look forward to receiving your revised manuscript.

Kind regards,

Alireza Soltani

Darrell A. Worthy, Ph.D

Academic Editors

PLOS ONE

Journal Requirements:

2) Your ethics statement must appear in the Methods section of your manuscript. If your ethics statement is written in any section besides the Methods, please move it to the Methods section and delete it from any other section. Please also ensure that your ethics statement is included in your manuscript, as the ethics section of your online submission will not be published alongside your manuscript.

Additional Editor Comments (if provided):

Reviewers' comments:

Reviewer's Responses to Questions

**Comments to the Author**

1. If the authors have adequately addressed your comments raised in a previous round of review and you feel that this manuscript is now acceptable for publication, you may indicate that here to bypass the “Comments to the Author” section, enter your conflict of interest statement in the “Confidential to Editor” section, and submit your "Accept" recommendation.

Reviewer #1: (No Response)

2. Is the manuscript technically sound, and do the data support the conclusions?

Reviewer #1: Partly

3. Has the statistical analysis been performed appropriately and rigorously? 

Reviewer #1: Yes

4. Have the authors made all data underlying the findings in their manuscript fully available?

Reviewer #1: No

5. Is the manuscript presented in an intelligible fashion and written in standard English?

Reviewer #1: Yes

6. Review Comments to the Author

Reviewer #1: The authors were able to address some, but not all, issues raised in prior reviews. Further clarification shed light on other issues that need to be addressed.

1. The authors suggest that change in value occurs during decision-making. They state in their abstract “…that CIPC is at least partially- due to the refinement of option value representations that occurs while people are pondering about choice options. For example, contemplating competing possibilities during a choice may highlight aspects of the alternative options that were not considered before. In the context of difficult decisions, this would enable people to reassess option values until they reach a satisfactory level of confidence.” And then later: “Critically for our theory, the more difficult the decision, the more deliberation and potential reassessment of value representations, the more likely a change of mind and the related CIPC.” And then after that: “Recall that, under our hypothesis, the deliberation that takes place during the decision process is expected to cause a refinement of internal value representations until a target level of choice confidence is met and the decider commits to a choice.” Similar statements are made at various other points in the manuscript.

As noted in both prior reviews – if reassessing values during decision-making is a critical feature driving CIPC, then response time is an important data point that could corroborate the authors’ assumptions. Namely – the more people are “contemplating competing possibility”, the longer the decision should take to make (i.e., longer the RT). This should, in turn relate to change in values. Failure to report RT because it was not collected is at minimum a critical limitation of the study. If RT was collected and does not relate to change in value, this is an important point that must be discussed as it is inconsistent with the explanation the authors describe throughout the manuscript. I disagree with the authors’ conclusion that RT is irrelevant to supporting the contentions they explicitly note above and throughout the manuscript. If the authors have misspoken throughout the manuscript about the cognitive mechanisms they believe are implicated in CIPC, then those inaccuracies need to be corrected.

2. Grammar problems in this sentence: “In brief, post-choice cognitive dissonance reduction theory (hereafter: post-choice CDRT) states that people come to liking what they have chosen. We rather suggest, somewhat trivially, that they may simply be choosing what they have come to liking.”

3. I appreciate the inclusion of the new figure 1. Unfortunately, it was not as helpful as it could have been. This may be due to insufficient description/labeling of the Y axis (is it appropriate to conceptualize this more simply as partial correlation with SoA?) and lack of clarity about what value certainty and choice confidence represent (selected – unselected items; unselected – selected; something else?).

4. It’s great that the authors collected more data to deal with the apparent age difference across groups. I also appreciate that the authors provided statistics about sex and age differences across conditions – I am not entirely clear on why they deemed such a request as unfair as it is standard practice in most journals to report statistics on demographic features of samples. Because p = .06 is still a marginal difference, this issue should be noted as a limitation in the discussion section. Additionally, please demonstrate that results are significant after controlling for age.

5. Given that the authors are testing competing models that have different predicted directionality (positive vs. negative correlations, if Figure 1 is in fact depicting partial correlations with SoA), the use of one-sided tests in some of their most critical analyses are not appropriate. For example, results from a one-sided analysis are purported to demonstrate: “The latter findings support our hypothesis, and are incompatible with classical post-choice CDRT.”

6. “We then asked whether post-choice ratings better predict choice (and choice confidence) than pre-choice ratings. First, we performed another logistic regression of paired choices, this time against the difference in post-choice value ratings (ratings #2). The ensuing choice prediction accuracy is higher than with pre-choice value ratings (accuracy = 0.787, 95% CI [0.770, 0.804], accuracy gain = 0.103, 95% CI [0.082, 0.124], p<0.001) (Figure 3).” The second sentence does not mention anything about pre-choice ratings. Clarify. Same comment on the next analysis described in the “Second, we regressed choice confidence…” part.

7. The traditional result is that the difference in value for two items in a pair increases when the pair are similar in pre-choice value. As described in the results section: “ "spreading of alternatives" (SoA), calculated as the mean difference in value rating gains between chosen and unchose items (SoA = [rating#2-rating#1]chosen - [rating#2-rating#1]unchosen).” It is unclear if this is the effect described here: “As expected, a random effect analysis on the ensuing parameter estimates shows that SoA significantly decreases with the absolute difference in pre-choice value ratings.” Is it that SoA decreases as the absolute difference in pre-choice value ratings increase? Likewise, description of directionality is needed for results that describe SoA in relation to pre-choice value certainty ratings (i.e., SoA decreases when pre-choice value certainty ratings are higher/lower) and choice confidence (SoA increases with higher/lower levels of choice confidence).

7. PLOS authors have the option to publish the peer review history of their article (what does this mean?). If published, this will include your full peer review and any attached files.

Reviewer #1: No

---

## [Author Response · Author response to Decision Letter 2]

21 Feb 2020

[We provide a point-by-point response to the reviewer in the attached latter]

---

## [Editor Report · Decision Letter 3]

28 Feb 2020

PONE-D-19-17905R3

Choosing what we like vs liking what we choose: How choice-induced preference change might actually be instrumental to decision-making

PLOS ONE

Dear Dr. Daunizeau,

Another Associate Editor and I have now read your response to the final reviewer's criticisms/comments. Overall, we think that you have addressed all issues except one (about the relationship between RT and CIPC). As a result, we feel that your manuscript has merit but does not fully meet PLOS ONE’s publication criteria as it currently stands. Therefore, we invite you to submit a revised version of the manuscript that addresses this issue as described below.

Specifically, it is not clear how "efforts" is defined for making decisions and what does "enough efforts" mean.

It seems that more efforts is needed for difficult decisions, which should result in longer RT and larger or at least similar CIPC (if the efforts are not being wasted in distinguishing between the two options), which predict no relationship between RT and CIPC, instead of negative correlation as suggested in the manuscript.

Alternatively, the same amount of effort could mean the same amount of time or even less time on difficult decisions (because the rate of effort spent should be higher for difficult decisions), which predicts that difficult trials should be faster and thus more confident, as predicted by the negative correlation between RT and confidence (e.g., Kiani et al 2014; De Martino et al, 2013). In summary, the use of "effort" to explain the observed negative correlation with RT and CIPC does not seem well justified. Therefore, we suggest that instead of using this observation as an evidence supporting their hypothesis, the authors should accept their finding on the negative correlation between RT and CIPC as a piece of evidence that might not support their hypothesis.

We would appreciate receiving your revised manuscript by Apr 13 2020 11:59PM. To enhance the reproducibility of your results, we recommend that if applicable you deposit your laboratory protocols in protocols.io, where a protocol can be assigned its own identifier (DOI) such that it can be cited independently in the future. For instructions see: http://journals.plos.org/plosone/s/submission-guidelines#loc-laboratory-protocols

We look forward to receiving your revised manuscript.

Kind regards,

Alireza Soltani

Academic Editor

PLOS ONE

---

## [Editor Report · Decision Letter 4]

17 Mar 2020

Choosing what we like vs liking what we choose: How choice-induced preference change might actually be instrumental to decision-making

PONE-D-19-17905R4

Dear Dr. Daunizeau,

We are pleased to inform you that after a discussion between editors, your manuscript has been judged scientifically suitable for publication and will be formally accepted for publication once it complies with all outstanding technical requirements, **and fulfills our final editorial request below.**

**Specifically, we found the discussion of how the DDM with collapsing bounds can account for a different relationship between CIPC and RT very useful for the readers. Therefore, we would like to ask you to include a more detailed explanation of that in the Discussion and include your "A note on DDM with collapsing bounds" as a supplemental material. We did not want burden you and/or delay the publication with another round of "minor revision" assuming that you will fulfill our request in your final submission.**

With kind regards,

Alireza Soltani

Academic Editor

PLOS ONE
---

## [Editor Report · Acceptance letter]

6 Apr 2020

PONE-D-19-17905R4 

Choosing what we like vs liking what we choose: How choice-induced preference change might actually be instrumental to decision-making. 

Dear Dr. Daunizeau:

I am pleased to inform you that your manuscript has been deemed suitable for publication in PLOS ONE. Congratulations! Your manuscript is now with our production department. 

With kind regards,

on behalf of

Dr. Alireza Soltani 

Academic Editor

PLOS ONE